# Co-Treatments of Gardeniae Fructus and Silymarin Ameliorates Excessive Oxidative Stress-Driven Liver Fibrosis by Regulation of Hepatic Sirtuin1 Activities Using Thioacetamide-Induced Mice Model

**DOI:** 10.3390/antiox12010097

**Published:** 2022-12-30

**Authors:** Jin A Lee, Mi-Rae Shin, JeongWon Choi, MinJu Kim, Hae-Jin Park, Seong-Soo Roh

**Affiliations:** 1Department of Herbology, College of Korean Medicine, Daegu Haany University, Daegu 42158, Republic of Korea; 2Research Center for Herbal Convergence on Liver Disease, Daegu Haany University, Gyeongsan 38610, Republic of Korea; 3Bio Convergence Testing Center, Daegu Haany University, Gyeongsan 38610, Republic of Korea

**Keywords:** Gardeniae Fructus, silymarin, Sirtuin 1, oxidative stress, liver fibrosis

## Abstract

*Gardeniae Fructus* (GF, the dried ripe fruits of *Gardenia jasminoides* Ellis) has traditionally been used to treat various diseases in East Asian countries, such as liver disease. Silymarin is a well-known medicine used to treat numerous liver diseases globally. The present study was purposed to evaluate the synergistic effects of GF and silymarin on the thioacetamide (TAA)-induced liver fibrosis of a mouse model. Mice were orally administered with distilled water, GF (100 mg/kg, GF 100), silymarin (100 mg/kg, Sily 100), and GF and silymarin mixtures (50 and 100 mg/kg, GS 50 and 100). The GS group showed remarkable amelioration of liver injury in the serum levels and histopathology by observing the inflamed cell infiltrations and decreases in necrotic bodies through the liver tissue. TAA caused liver tissue oxidation, which was evidenced by the abnormal statuses of lipid peroxidation and deteriorations in the total glutathione in the hepatic protein levels; moreover, the immunohistochemistry supported the increases in the positive signals against 4-hydroxyneal and 8-OHdG through the liver tissue. These alterations corresponded well to hepatic inflammation by an increase in F4/80 positive cells and increases in pro-inflammatory cytokines in the hepatic protein levels; however, administration with GS, especially the high dose group, not only remarkably reduced oxidative stress and DNA damage in the liver cells but also considerably diminished pro-inflammatory cytokines, which were driven by Kupffer cell activations, as compared with each of the single treatment groups. The pharmacological properties of GS prolonged liver fibrosis by the amelioration of hepatic stellate cells’ (HSCs’) activation that is dominantly expressed by huge extracellular matrix (ECM) molecules including α-smooth muscle actin, and collagen type1 and 3, respectively. We further figured out that GS ameliorated HSCs activated by the regulation of Sirtuin 1 (Sirt1) activities in the hepatic protein levels, and this finding excellently reenacted the transforming growth factor-β-treated LX-2-cells-induced cell death signals depending on the Sirt1 activities. Future studies need to reveal the pharmacological roles of GS on the specific cell types during the liver fibrosis condition.

## 1. Introduction

Chronic liver disease (CLD), which causes approximately 2 million deaths every year, is one of the major medical concerns in the world [1]. As an end stage of CLD, liver cirrhosis is the 11th most common cause of death in the world [2]. The global burden to treat or cure liver cirrhosis has steadily increased annually; however, no effective therapeutics are available excluding liver transplants [3]. Liver fibrosis, which is an early stage of liver cirrhosis, is reversible to the normal status of the liver, thus this stage is very important to decide whether to progress liver cirrhosis or reverse to the normal status [4,5]. Hence, many studies have focused on liver fibrosis to develop its protective and treatment agents. An evidence accumulation has provided good indications that hepatic stellate cells (HSCs) play central roles in the progression of liver fibrosis [6]. During liver fibrosis owing to various stimuli, HSCs are activated, especially in the conditions of fibroblast to myofibroblast, and this status leads to the production and accumulation of the excessive extracellular matrix (ECM) [7,8]. Accordingly, inhibition or regression of the HSCs’ activation may be a strategy for the treatment of liver fibrosis. Thus, recently, scavenging the activated HSCs has been considered as a new approach to reverse fibrogenesis [9,10]. However, despite these efforts, understanding of the exact pathological mechanism of liver fibrosis is still poorly described.

On the other hand, as alternative medicines, herbal plants or their derived natural products have actively been recommended to treat CLD, especially focusing on liver fibrosis using cell lines or primary cell-based in vitro experiments, animal models, and even clinical trials with scientific evidence [11,12,13]. Gardeniae Fructus (GF, the dried ripe fruits of *Gardenia jasminoides* Ellis) has traditionally been used to treat various diseases including homeostatic, antiphlogistic, analgesic, and antipyretic effects [14]. Recent studies have documented well that GF improves liver disease including CLD and non-alcoholic fatty liver diseases (NAFLD), with anti-inflammatory effects and the regulation of lipid metabolism [15,16]. Geniposide, a major compound of GF, is well known to possess major pharmacological activities as antioxidants and anti-inflammatories [17,18]. Silymarin extracted from the fruit and seeds of milk thistle exerts hepatoprotective effects via the enhancement of anti-inflammatory, antioxidative stress, and anti-cytotoxicity effects in various liver disease animal models and patients with CLD including fibrosis [19]. However, the effectiveness of silymarin is limited because of the insufficiency of aqueous solubility, the poverty intestinal absorption, and, likewise, the lack of systemic bioavailability when it is performed orally [20,21]. Accordingly, a range of strategies associated with the formulation of silymarin have been studied actively for its enhanced effect via the oral route [22].

Silencing information regulator 1 or Sirtuin 1 (Sirt1), is a lysine deacetylase family member, sharing significant homologies with the yeast protein Sir2 [23] and previous evidence suggests a major role for Sirt1 in the metabolic regulation of liver injury. Additionally, our previous study confirmed the anti-hepatofibrotic effects of GF and silymarin via the regulation of hepatic Sirt1 activities using a TAA-induced liver fibrosis model, especially ameliorations of pro-fibrogenic cytokines, decreases in ECMs, and increases in antioxidant effects by regulation of the LKB1/AMPKα/NF-κB signaling pathway [24]. The latest research has shown that combination treatment could have an advantage in the treatment of metabolic diseases such as obesity, diabetes, and atherosclerosis, or virus diseases such as AIDS [25]. In particular, the synergy effect of combination therapy is generally referred to as drug–drug interaction and is used to produce combined effects that are better than the individual effects of each of them. A well-known case of the combined effects is the co-treatment of metformin and aspirin in cancer [26]. Accordingly, the above reasons led us to set up a possible hypothesis that co-treatments with GF and silymarin (GS) may synergistically alleviate liver fibrosis with unique pharmacological actions as compared with each of the single treatments. Moreover, it is considered that the above-mentioned limitations of using silymarin alone can be overcome with this co-treatment therapy.

Thus, herein, we aimed to investigate the synergistic effects of GF and silymarin in a concomitant manner on TAA-induced liver fibrosis using a mouse model by focusing on the antioxidant and anti-fibrotic effects as compared with each of the single treatment groups. Through the current study, we found the novel synergistic functions of GS are a potential therapeutic candidate via the scavenging of activated HSCs during the completion of fibrotic alteration.

## 2. Material and Methods

### 2.1. Preparation of the Plant Material and Finger Printing Analysis of GF

The GF was produced in Goheung, Jellannam-do in 2019 and was supplied from Bonchowon (Yeongcheon, Gyeongsanbuk-do, Republic of Korea). The GF (100 g) was extracted by 10× volume of distilled water (100 °C) for 2 h. As the result of finger printing analysis, the content of geniposide, which is the active ingredient of GF, was analyzed to be 176.625 ± 0.809 (mg/g) [24].

### 2.2. Mice Treatment

The animal experiment was approved by the Institutional Animal Care and Use Committee of Daegu Haany University and performed according to the Guidelines for Animal Experiment (approval no. DHU2021-021). C57BL/6 mice (male, 20–25 g) were purchased from DBL (Eumseong, Republic of Korea). After 1 week of adaptation (environmental conditions: 12 h light/dark cycle, controlled temperature (22 ± 2 °C), and humidity (50 ± 5%)), mice were divided into 6 groups (*n* = 8 for each group): Control, TAA (TAA only), GF 100 (100 mg/kg), and Sily 100 (100 mg/kg), GS 50, and 100 (50 and 100 mg/kg), respectively. Mice in the control group were intraperitoneally (i.p.) injected 0.9% normal saline orally administrated with distilled water (DW); the TAA group received TAA (i.p.) and DW (peroral, p.o.); the GF 100 group received TAA (i.p.) and GF at 100 mg/kg/day (p.o.); the Sily group received TAA (i.p.) and silymarin at 100 mg/kg/day (p.o.); the GS 50 and 100 group received TAA (i.p.) and GS at 50 and 100 mg/kg/day (p.o.) for 8 weeks of experimental periods. Liver fibrosis was induced by TAA three times injection per week for 8 weeks according to an escalating treatment dose protocol (100 mg/kg at 1 week, 200 mg/kg at 2–3 weeks, and 400 mg/kg at 4–8 weeks, respectively). The treatment drugs for liver fibrosis both GF (100 mg/kg/day), silymarin (100 mg/kg/day), and GS (50 and 100 mg/kg/day) were administrated 90 min prior to TAA injection. Serum was obtained by centrifugation of blood for 10 min (at 4000 rpm, 4 °C), and stored in −80 °C freezer for subsequent biochemical evaluation. Liver tissues were transferred 10% neutral formalin for the purpose of histological analysis and stored at −80 °C for further biochemistry analysis.

### 2.3. Histological Examination

Liver tissues were fixed with 10% formalin and then embedded and sectioned. These were sectioned with 3 μm thickness. Liver tissue sections were stained with three of the most suitable dyes, such as hematoxylin and eosin (H&E), Masson’s trichrome (MT), and Sirius Red, which was followed by the standard protocol. The images were captured using Olympus BX51 Microscope (Olympus Co., Ltd., Tokyo, Japan).

### 2.4. Analysis of Serum Biochemistry

Liver enzymes such as aspartate aminotransferase (AST) and alanine aminotransferase (ALT) were measured by serum levels using a Transaminase CⅡ-Test (Wako Pure Chemical Industries Ltd., Osaka, Japan). Serum lactate dehydrogenase (LDH) (Sigma Aldrich Co., St Louis, MO, USA) and ammonia (Abcam, Cambridge, UK) were measured by ELISA kit according to the manufacturer’s instructions.

### 2.5. Cell Death Signaling

Cell death signal analysis was performed using the TUNEL assay kit and was analyzed according to the manufacturer’s instructions (ApopTag Fluorescein Direct In Situ Apoptosis Detection Kit, S7100, Millipore, Burlington, MA, USA).

### 2.6. Analysis of Immunohistochemistry (IHC)

Oxidative stress (4-HNE), DNA damage (8-OHdG), Kupffer cells (F4/80), α-smooth muscle actin (ASMA), collagen type 1, and collagen type 3 were detected by IHC analysis. IHC analysis was carried out in the same way as in the previous study [24]. The images of IHC analysis were captured using light microscopy (Olympus BX51, Tokyo, Japan).

### 2.7. Measurement of ROS, TBA-Reactive Substance (TBARS) Levels and Pro-Inflammatory Cytokine

TBARS assay to evaluate malondialdehyde (MDA) was estimated using commercially available kit (BioAssay System, Hayward, CA, USA). Hepatic ROS levels were measured using DCF_DA fluorescence probe. Briefly, small pieces of hepatic tissue were homogenized in 10 mM PBS (pH.7.2) and centrifuged at 3000× *g*, 20 min, 4 °C. Then supernatants were collected and transferred 100 µL of sample lysates to black wall of 96-well microplate and 10 µL of DCF_DA 10 µM were added to the plate. Plate was incubated in 37 °C for 20 min and fluorescents were measured under the excitation/emission at 485 nm/535 nm of wavelengths. Results appeared as fold changes, which were normalized by control group. Hepatic protein levels of pro-inflammatory cytokines were measured using commercially available kits according to the manufacturer’s protocol (mouse TNF-α and mouse IL-6 for BD Bioscience, San Jose, CA, USA; mouse IL-1β for R&D system, Minneapolis, MN, USA).

### 2.8. Cell Cultures

HepG2 cells (human blastoma cell line) and LX-2 cells (hepatic stellate cell line) were used in the experiments. The LX-2 (immortalized human HSCs) cell line was a kind gift from Dr S. L. Friedmann (Icahn School of Medicine at Mount Sinai, New York, NY, USA) and HepG2 cells were supplied by American Type Culture Collection (Manassas, VA, USA). HepG2 cells were cultured in DMEM-supplemented 10% fetal bovine serum (FBS) and LX-2 cells were cultured in DMEM-supplemented 5% FBS (37 °C incubators with 5% CO_2_).

For hepatic oxidative stress condition, it was pretreated with GF, silymarin, and GS then treated with 500 µM of hydrogen peroxide (H_2_O_2_) for 6 h. Cells were washed with 10 mM PBS (pH 7.3) twice and fixed in 4% paraformaldehyde (PFA) or cell lysis buffers.

For liver fibrosis condition, it was pretreated with GF, silymarin, and GS for 6 h. After, LX-2 cells were activated by transforming growth factor (TGF)–β1 (10 ng/mL) for 18 h. IHC analysis using HepG2 or LX-2 cells was performed in the same manner as in previous studies according to standard protocols [24].

For cell death signaling, commercial kits (Dead Cell Apoptosis Kits with Annexin V for flow cytometry, V13242, Thermo Fishers, Waltham, MA, USA) were used and analyzed by BD FACSDiva™.

### 2.9. Western Blotting

To obtain tissue samples, we performed them in the same way as in previous studies [24]. Samples containing 10 μg of protein were electrophoresed through 8–15% SDS-PAGE and transferred to a nitrocellulose membrane. Primary and secondary antibodies were incubated on each membrane and then visualized using enhanced chemiluminescence (ECL) reagent (Cyanagen Srl, Bologna, Italy). Bands were detected using Sensi-Q 2000 Chemidoc (Lugen Sci Co., Ltd., Gyeonggi-do, Bucheon-si, Korea) and quantified through Image J 1.52 software (NIH, Bethesda, MD, USA).

### 2.10. Statistical Analysis

All data were expressed as mean ± SEM by the performance of the Prism 9.2 software from GraphPad (La Jolla, CA, USA). Comparisons between the two groups were performed using a two-tailed unpaired Student *t*-test. Moreover, for comparisons for more than two groups, we performed one-way or two-way ANOVA followed by Tukey’s post hoc tests.

## 3. Results

### 3.1. Co-Treatments of GF and Silymarin Ameliorate TAA-Injected Hepatic Injury

Repeated injection of TAA caused significant increases in inflamed cell infiltrations and necrotic areas through the liver tissues by inspection of the histopathological analysis of H&E staining (Figure 1A), whereas GS treatment markedly exerted diminishment of them. These pathological alterations were well supported by the abnormal status of serum liver enzymes including 16.5-, 5.5-, 3.2-, and 3.5-fold increases in ALT, AST, LDH, and ammonia levels, respectively, as compared with the control group (*p* < 0.01 or 0.001 in Figure 1B–E). Administrations with GS considerably decreased serum levels of hepatic injury as compared with the TAA group (*p* < 0.01 or 0.001), and a high dose of GS significantly decreased serum levels compared with each single treatment group (*p* < 0.05 or 0.01).

### 3.2. Synergistic Effects of GS on the Liver Cell Deaths by Attenuations of Oxidative Stress

We further observed the cell death signals by application of the TUNEL assay. As we expected, TAA injection led to the enhancement of liver cell deaths in the liver tissues as compared with the control group; however, GS treatments remarkably reduced these signals of the TUNEL assay through the liver tissues (Figure 2A). Increased cell death induced by TAA was significantly decreased in the GS treatment than in the GF single treatment (*p* < 0.01 or 0.001 in Figure 2B).

To verify the etiology of liver cell deaths during TAA injection of liver fibrosis, next, we investigated the oxidative stress-related molecules. As expected, hepatic ROS levels in the TAA group significantly increased as compared with the control group, whereas GS administrations synergistically decreased these alterations as compared with the GF 100 group (*p* < 0.01 and 0.05 for GS 50 and GS 100 in Figure 3A), respectively. Lipid peroxidations, which are the final products of oxidative stress, in the serum and hepatic protein levels were drastically elevated by TAA treatment as compared with the control group, respectively (*p* < 0.01 in Figure 3B,C); however, treatments with a high dose of GS significantly decreased them and they also exhibited remarkable effects as compared with each single treatment group (*p* < 0.05 in Figure 3B,C). IHC analysis by targeting against 4-HNE and 8-OHdG displayed that these oxidative stresses contributed well to hepatic injuries due to TAA injection, whereas GS groups notably decreased them (Figure 3D,E). Furthermore, antioxidant components in the liver protein levels of total GSH contents and SOD activities were reduced by TAA injection, while administrations with both GF and Sily significantly increased them and GS showed, synergistically, exertions of both antioxidant components compared with silymarin 100 mg/kg treatment (Appendix A
*p* < 0.05, 0.01, and 0.001 relying on the dose-dependent manner).

### 3.3. Anti-Inflammatory Effects of GS in the Liver Tissue

We evaluated the hepatic inflammation by focusing on the hepatic-resided macrophage markers, which are known as Kupffer cells (KCs), by Western blot analysis. Hepatic protein levels of p-NFκB in the TAA group were remarkably increased as compared to the control group, whereas IκBα levels were decreased. Administration with GS 50 and 100 exerted recovery of the above alterations (Figure 4A). In particular, KC activation-related markers such as iNOS and TLR were increased by TAA injection, but they were effectively decreased in GS-treated groups (Figure 4A). The above alterations coincided well with the IHC analysis against F4/80 staining (Figure 4B). Hepatic protein levels of pro-inflammatory cytokines including TNF-α, IL-1β, and IL-6 showed the anti-hepatic inflammation effects of GS on TAA-induced liver injury conditions (*p* < 0.05 and 0.01, in Figure 4C–E) but not the synergistic effects of GS compared with each single treatment group.

### 3.4. Anti-Liver Fibrosis Effects of GS on the Liver Tissue with Synergistic Manner

Our outcomes clearly exhibited the synergistic effects of GS on liver fibrosis as compared with each single treatment group. Trichrome staining, which is a marker of collagen deposition through liver tissues, showed that, as expected, GF100, Sily100, GS50, and GS100 significantly reduced collagens as compared with the TAA group (*p* < 0.05 or 0.01 in Figure 5A and Appendix A), and it is also showed that the GS100 group significantly decreased collagen depositions as compared with the GF100 group (*p* < 0.05 in Figure 5A and Appendix A). Moreover, ECM molecules including ASMA, ColT1, and ColT3, by evidence of IHC analysis, displayed the synergistic properties of GS better than other groups in a dose-dependent manner as compared with each single treatment group (*p* < 0.05 in Figure 5B–D and Appendix A), respectively. Western blot analysis data clearly provided the effects of GS against hepatic fibrotic molecules including ECMs (Acta2, ColT1, and ColT3), pro-fibrogenic cytokines receptors including TGF-β1R and PDGFβR, and their regulators of Smad2 and Smad3 activation, and modulators of ECM synthesis proteins, especially for TIMP1 and MMP13 with a synergistic manner, respectively (Figure 5E). Sirius Red staining supports the above results that our drug treatments significantly decreased collagen accumulations (Appendix A), and GS showed synergetic effects as compared with the GF 100 (*p* < 0.01).

### 3.5. Pharmacological Actions of GS through Nrf2/HO-1 Signaling Pathway via Regulations of Hepatic Sirt1 Activation

In the liver tissue, we further investigated the pharmacological actions of GS to explain the underlying mechanism of GS as compared with each single treatment group. Hepatic protein levels of Nrf2 were remarkably decreased in the TAA group and the Sily 100 group led to the recovery or prevention of the depletion of Nrf2 as compared with the TAA group. Both the GS 50 and 100 groups showed relatively increased Nrf2 in the hepatic protein levels as compared with the GF100, but not the Sily100 group (Figure 6A). HO-1 also supported the alterations in Nrf2 in the hepatic protein levels well (Figure 6A). We found AMPKα activation was drastically decreased in the TAA group; interestingly, both the GF and Sily 100 groups did not recover its activation but both doses of GS treatments notably recovered it (Figure 6A). Figure 6B shows the above alterations would be regulated by hepatic Sirt1 activations mainly by the synergistic manners of GS50.

### 3.6. Selective Effects of Sirt6 by Induction of HSCs during Fibrosis Development

To investigate the possible mechanisms of GS during liver fibrosis development, next, we performed a cell-line-based in vitro experiment using a human hepatoma cell line (HepG2 cells) and human HSCs (LX-2 cells), HepG2 and LX, respectively. As expected, GS synergically prevented cell death signals via the attenuation of oxidative stress, which was induced by H_2_O_2_ treatment, as compared with each single treatment of GF and silymarin (Appendix A). These antioxidant and anti-apoptosis effects were attributed by Sirt1, which was enhanced by GS pretreatment (Appendix A). Regarding the anti-liver fibrosis effects of Sirt6 on activated HSCs, GS predominantly prevented accumulations of collagen type 1 deposition during TGF-β1-mediated fibrotic stimulus on LX-2 cells (Figure 7A,B). Additionally, we clearly discovered the selective pharmacological actions of GS, which synergically induced LX-2 cell deaths, especially the enhancement of apoptosis, by using the TUNEL assay and flow cytometry results (Figure 7C; Appendix A).

## 4. Discussion

In the present study, we aimed to investigate the synergistic effects of co-treatments with GF and silymarin on TAA-induced liver fibrosis using a mouse model. We hypothesized as follows: (1) GF and silymarin co-treatment would synergistically work on liver fibrosis, (2) if so, what kind of condition would be more dominantly acted on as part of the implication during liver fibrosis, and (3) which molecular signaling would be applied to GS as compared with a single treatment of GF or silymarin.

First, we found the effects of GS on the liver injury markers including histopathological alterations as well as serum biochemistries. Inflamed cell infiltrations were significantly reduced through the liver tissue as compared with the TAA group in the GS group; these properties were well supported by the serum levels of liver injury by the synergistic effects of GS as compared with the GF100 or Sily100 groups, respectively. These results could anticipate what kind of pathological condition may be implicated in the pharmacological effects of GS against TAA-induced liver fibrosis.

TAA is well known for hepatotoxin, and it is attributed to the hepatocyte deaths such as apoptosis and necroptosis [27,28]. Our IHC analysis also showed the synergistic effects of GS on the hepatocyte deaths by TUNEL assay compared with the GF treatment but not silymarin administration. The previous study figured out the toxic effects of TAA in the hepatic tissue [29]. TAA is mainly metabolized by CYP2E1, then it converts to thioacetamide sulfoxide, which is a highly reactive molecule, and it further undergoes another metabolism process with thioacetamide-S, S-dioxide [30,31]. These processes trigger oxidative stress in the liver tissue. Thus, we next focused on the oxidative stress and antioxidant components. According to our expectations, GF, Sily, and GS treatments remarkably decreased hepatic ROS, serum MDA, and hepatic protein MDA levels. IHC against both 4-HNE (a marker of lipid peroxidation) and 8-OHdG (a marker of DNA damage), and hepatic protein levels of total GSH contents and SOD activities also supported these antioxidant effects of each drug treatment. Interestingly, we found that similar to the hepatocyte deaths, GS showed synergistic effects mainly occurred when compared with the Sily100 group. This condition led to a focus on another possible pathological condition, especially inflammation.

Oxidative stress in the liver tissue generally accompanies inflammation. Furthermore, TAA participates to release iNOS and activates NFκB, which are mainly from KCs and lead to hepatocyte deaths by the secretion of pro-inflammatory cytokines as a result of the activation of KCs [32,33]. Thus, we observed that hepatic KCs through the liver tissue by applications of F4/80 and GS, show the tendency of a reduction in the numbers of, and a decrease in the KCs’ activation-related proteins including iNOS, NFκB, and TLR4 (Figure 4A,B), which coincided well with pro-inflammatory cytokines, but, interestingly, GS did not show synergistic effects. Since GS did not show its synergistic properties on the hepatic inflammation, we determined the haptic fibrosis signaling pathways by focusing on the ECM’s production and pro-fibrogenic cytokine proteins.

The exact pathophysiological mechanisms of liver fibrosis remain unclear, but it is a physiological progression of wound and healing. Under this process, HSCs play their roles to precede fibrosis alteration. The activation of HSCs, especially the progression of fibroblasts to myofibroblasts, is the main target that leads to the deposition of ECMs by the release of pro-fibrogenic cytokines such as TGF-β and PDGB-β [34,35,36]. Additionally, liver fibrosis is the stage of the reversal to normal liver tissue [7,8,36,37,38]; however, there is no therapeutic access to these critical stages. In the present study, we found out that all of the drug treatments, GF, silymarin, and their co-treatments, work efficiently to significantly reduce collagen deposition and ECMs through liver tissues. Additionally, Western blot analysis supported these properties in hepatic protein levels, especially by modulations of the TGF-β1/Samd3-signaling pathway. Our data also showed the synergistic effects of GS as compared with other single treatments that caused ameliorations of fibrotic molecules and ECM accumulations.

To understand the underlying actions of GS, we applied possible molecular signaling pathways. Our previous study evidenced well the anti-liver fibrosis effects of GF on the TAA-induced liver fibrosis model via the enhancement of hepatic Sirt1 activation [24]. The co-treatments of GF and silymarin also showed the enhancement of hepatic Sirt1 activity by Western blot analysis. Furthermore, the antioxidant key modulator such as the Nrf2/HO-1 signaling pathway was dominantly mediated by GS treatment as compared with other single treatments. Sirt1 is known for its modification of the histone and non-histone protein, and it is also considered a cellular metabolic sensor by its ability to sense the metabolic status of the cell to the chromatin structure [39,40]. In the liver tissue, recent studies have verified the beneficial roles of Sirt1 in various liver diseases including alcoholic liver diseases, non-alcoholic liver diseases, and cholestatic liver injury using animal models with the regulation of lipid metabolism (PPAR-α), antioxidant components (such as HO-1), and cell-death-related proteins or genes (p53 or Foxo1a) [41,42,43,44]. Additionally, Sirt6 deeply implicates the regulation of redox homeostasis, inflammation, and aging [45,46,47,48]. Previous studies have reported the role of Sirt1 in liver fibrosis by the inhibition of activated HSCs through its unique role of histone demethylation or histone deacetylation by targeting PPAR in HSCs [49,50]. The effects of scavenges of the activated HSCs under the liver fibrosis condition have not been revealed yet. Thus, next, our strategy of GS treatment was focused on the induction of cell deaths against activated HSCs using TGF-β1-treated LX-2 cells. Our findings addressed the effects of GS on the liver cell type-specific roles of Sirt1 in hepatocytes and HSCs, respectively. Interestingly, GS worked efficiently to survive hepatocytes under severe oxidative stress conditions by the enhancement of Sirt1 activities; however, Sirt1 markedly reduced collagen type1 production in activated HSCs through the enhancement of apoptosis signals during fibrotic stimulus.

From our study, we found the synergistic effects of GS in liver fibrosis as compared with other single treatments. As expected, liver injury due to TAA injection was considerably attenuated by GS, and it showed synergistic effects. The underlying mechanisms of GS may be involved in liver injury by oxidative stress-evoked hepatocyte death and the inhibition of HSCs’ activation, but do not actively work in inflammatory reactions. In the liver tissue, the main actions of GS are related to the antioxidant effects via regulation of the Nrf2/HO-1 signaling pathway, which was affected by hepatic Sirt1 activities. Interestingly we found out that GS modulates Sirt1 in the liver tissue during TAA-induced liver fibrosis, mainly controlling oxidative-stress-mediated hepatocyte deaths and liver fibrosis by the inhibition of HSCs’ activation, respectively. In the present study, we explored the unique role of GS, which could be a modulator of Sirt1 activities, using two different human liver cell lines, such as the hepatoma cell line and HSCs cell line. During oxidative stress conditions in hepatocytes, GS showed synergistic effects by the attenuation of oxidative stress as well as the inhibition of cell death signals. In HSCs for fibrotic status, GS induced the cell deaths of the activated HSCs by the regulation of Sirt1. These cell type-specific effects of GS may be contributing by preceding the resolution of liver fibrosis or preceding the prevention of fibrotic alterations during the fibrosis process in the liver tissue. Specific cell death induction in activated HSCs is a major key point to consider in anti-liver fibrosis therapies [51,52].

To avoid the progression of severe CLD conditions, such as cirrhosis or hepatocellular carcinoma (HCC), it is very important to consider the most adaptive medical care in the stage of liver fibrosis, especially focusing on the alleviation of activated HSCs with minimized harmful effects on hepatocytes during treatments. In our findings, administrations with GS, which is meant to give synergistic effects, showed outstanding effects on liver fibrosis in a mice model by selectively scavenging the activated HSCs compared with its single treatments. The limitations of this study include that it did not analyze the safety and toxicological issues based on the physiological levels. Thus, to apply clinical practice, the above issues need to be evaluated in the future. Additionally, further studies are also necessary to figure out the specific underlying actions and specific types of cell death in HSCs. Additionally, the combinations of the active compounds from GF and silymarin need to be explored for their effects on liver fibrosis conditions by the regulation of hepatic Sirt1 activities.

## Figures and Tables

**Figure 1 antioxidants-12-00097-f001:**
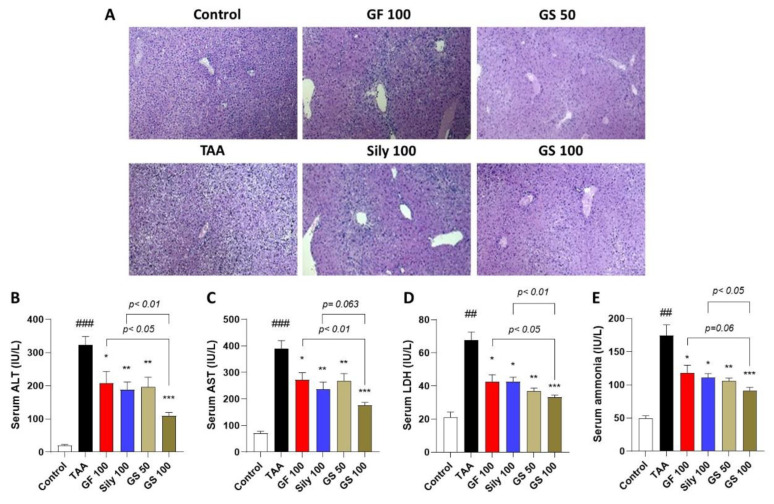
Co-treatments of GF and silymarin ameliorate TAA-injected hepatic injury. (**A**) Histopathological analysis of H&E staining. Serum levels of liver enzymes include ALT (**B**), AST (**C**), LDH (**D**), and ammonia (**E**). Data were expressed by mean ± S.E.M. ^##^
*p* < 0.01 and ^###^
*p* < 0.001 for Control vs. TAA. * *p* < 0.05, ** *p* < 0.01, and *** *p* < 0.001 for TAA vs. drug treatments (*n* = 3 for H&E staining, *n* = 6–8 for biochemistry analysis). Images were captured by light microscopy conditions (100× magnifications).

**Figure 2 antioxidants-12-00097-f002:**
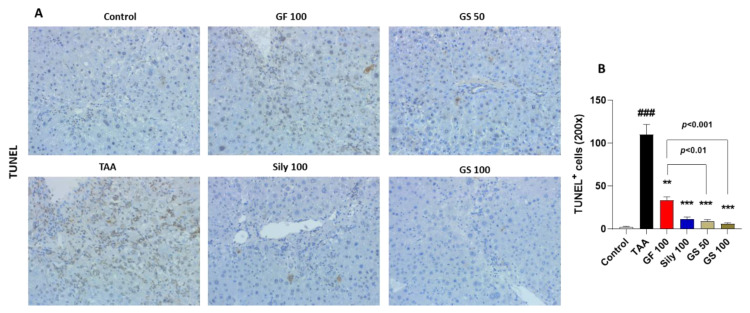
Synergistic effects of GS on the liver cell deaths by attenuations of oxidative stress. (**A**) TUNEL assay. (**B**) Quantification of TUNEL positive cells. Data were expressed by mean ± S.E.M. ^###^
*p* < 0.001 for Control vs. TAA. ** *p* < 0.01, and *** *p* < 0.001 for TAA vs. drug treatments (*n* = 4 for IHC analysis). Images were captured by light microscopy condition (200× magnifications).

**Figure 3 antioxidants-12-00097-f003:**
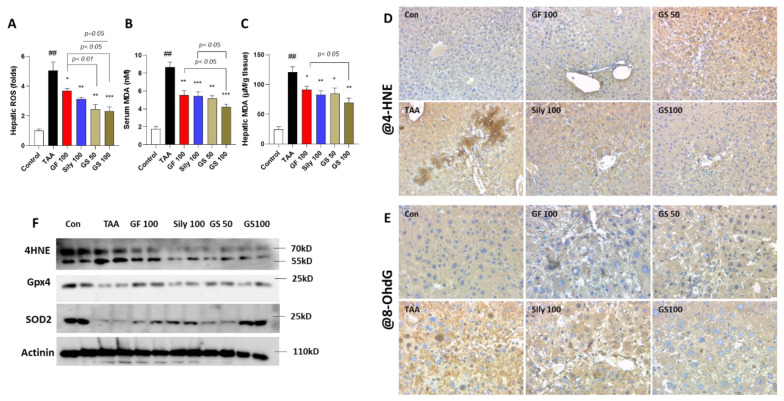
Administrations with GS exert amelioration of hepatic tissue oxidation. (**A**) Hepatic ROS levels. (**B**) Serum MDA levels. (**C**) Hepatic MDA levels. (**D**) IHC analysis against 4-HNE and (**E**) 8-OHdG. (**F**) Western blot analysis of the antioxidant-related proteins. Data were expressed by mean ± S.E.M. ^##^
*p* < 0.01 for Control vs. TAA. * *p* < 0.05, ** *p* < 0.01, and *** *p* < 0.001 for TAA vs. drug treatments (*n* = 4 for IHC analysis, *n* = 6–8 for biochemistry analysis, *n* = 4 for Western blot analysis). Images were captured by light microscopy conditions (100× magnifications).

**Figure 4 antioxidants-12-00097-f004:**
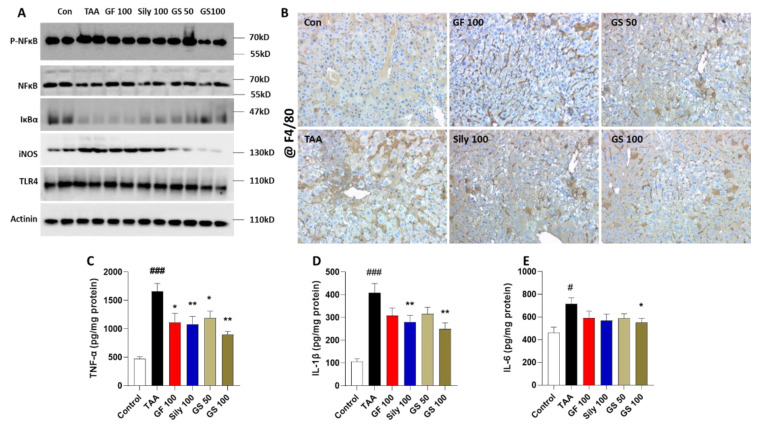
Anti-hepatic inflammation effects of GS by regulation of pro-inflammatory response. (**A**) Western blot analysis of the inflammation-related proteins. (**B**) Representative images of F4/80 in the liver tissue. (**C**) Hepatic protein levels of TNF-α, (**D**) IL-1β, and (**E**) IL-6. Data were expressed by mean ± S.E.M. ^#^
*p* < 0.05 and ^###^
*p* < 0.001 for Control vs. TAA. * *p* < 0.05 and ** *p* < 0.01 for TAA vs. drug treatments (*n* = 4 for IHC analysis, *n* = 6–8 for biochemistry analysis, *n* = 4 for Western blot analysis). Images were captured by light microscopy conditions (100× magnifications).

**Figure 5 antioxidants-12-00097-f005:**
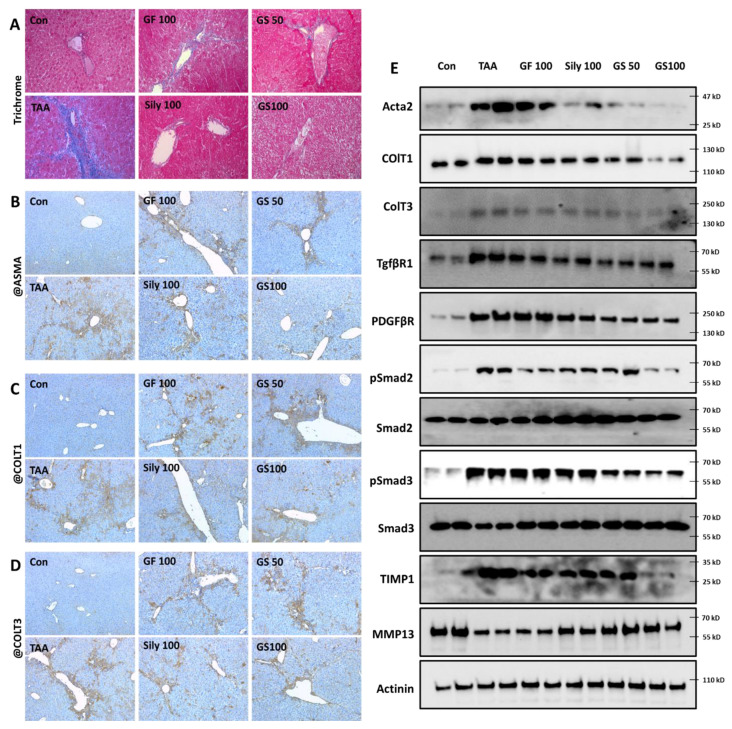
Anti-liver fibrosis effects of GS via attenuation of ECM production and pro-fibrogenic molecules. (**A**) Trichrome staining. IHC analysis against (**B**) ASMA, (**C**) ColT1 (collagen type1), and (**D**) ColT3 (collagen type3). (**E**) Western blot analysis of ECM production proteins and fibrogenic cytokine-related proteins (*n* = 4 for trichrome and IHC analysis, *n* = 4 for Western blot analysis). Images were captured by light microscopy conditions (200× for trichrome staining and 100× for IHC analysis).

**Figure 6 antioxidants-12-00097-f006:**
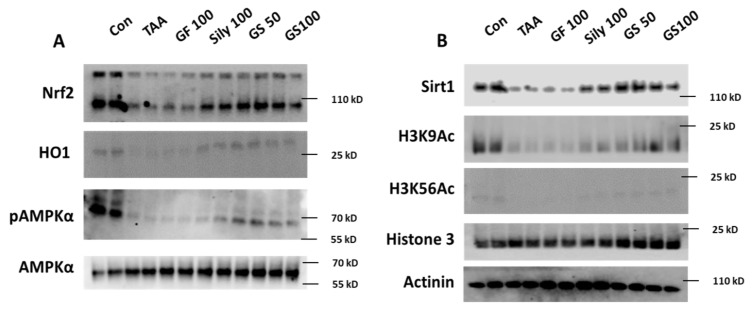
Pharmacological actions of GS through Nrf2/HO-1 signaling pathway via regulations of hepatic Sirt1 activation. Representative images of Western blot analysis (**A**) Nrf2/HO-1 signaling pathways, (**B**) Sirt1 activities in hepatic protein levels (*n* = 4 for Western blot analysis).

**Figure 7 antioxidants-12-00097-f007:**
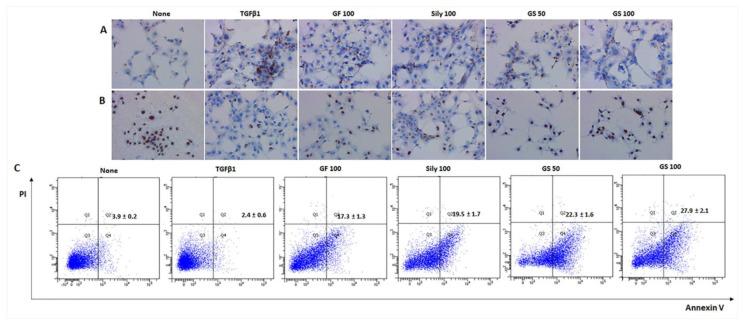
Selective effects of Sirt6 by induction of HSCs during fibrosis development. Representative images of IHC against (**A**) ColT1, (**B**) Sirt6. (**C**) Flow cytometry analysis. For HSCs’ activation, TGF-β1 (5 ng/mL) was treated after 6 h of drug treatments. Data were expressed by mean ± S.E.M. (*n* = 3 for IHC analysis, *n* = 4 for flow cytometry).

## Data Availability

The data presented in this study are available on request from the corresponding author.

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
