# Peer review of "Co-Treatments of Gardeniae Fructus and Silymarin Ameliorates Excessive Oxidative Stress-Driven Liver Fibrosis by Regulation of Hepatic Sirtuin1 Activities Using Thioacetamide-Induced Mice Model"

_antioxidants, 2022, doi:10.3390/antiox12010097_

Round 1
Reviewer 1 Report
The article „Co-treatments of Gardeniae Fructus and Silymarin Ameliorates Excessive Oxidative Stress Driven Liver Fibrosis by Regulation of Hepatic Sirtuin1 Activities using Thioacetamide-induced Mice model” evaluates the the synergistic effects of Gardeniae fructus and Silymarin on the rxprtimrntslly inducrd liver fibrosis on mice model of Thyoacetamide administration. The subject is interesting and of interest, in the area of the journal. The entire design is complex, well-described and performed
From my point of view, some issues should be clarified:
In the Abstract, the clear conclusion of the study should be added. The phrase „Admin[1]istrations with GS considerably the above alterations as compared with the control group, as well as its single high dose, treated groups” is incomplete.
In Introdution, a most clear aim of the study shoud be emphasized
In Discussion section, no Figure should be mentioned. This section must discuss the obtained results in relation with the findings from the literature. Limits and perspectives of the study should be emphasized. A clear conclusion shoud be stated.
Author Response
Comments and Suggestions for Authors
The article „Co-treatments of Gardeniae Fructus and Silymarin Ameliorates Excessive Oxidative Stress Driven Liver Fibrosis by Regulation of Hepatic Sirtuin1 Activities using Thioacetamide-induced Mice model” evaluates the the synergistic effects of Gardeniae fructus and Silymarin on the rxprtimrntslly inducrd liver fibrosis on mice model of Thyoacetamide administration. The subject is interesting and of interest, in the area of the journal. The entire design is complex, well-described and performed
From my point of view, some issues should be clarified:
1) In the Abstract, the clear conclusion of the study should be added. The phrase „Admin[1]istrations with GS considerably the above alterations as compared with the control group, as well as its single high dose, treated groups” is incomplete.
2) In Introdution, a most clear aim of the study shoud be emphasized
3) In Discussion section, no Figure should be mentioned. This section must discuss the obtained results in relation with the findings from the literature. 4) Limits and perspectives of the study should be emphasized. A clear conclusion shoud be stated.
â–º Thank you very much for the reviewer’s professional suggestion for improving the quality of the present manuscript. According to the reviewer’s critical comments, we re-wrote the above considerations on the ‘Revised Manuscript’. That is as follows.
1) We revised it to a full sentence.
2) The sentence below is added to further emphasize the purpose of this study in the Introduction section.
So, we added the references [20-23].
3) In the Discussion section, we described without mentioning the Figure and added the new Reference [29].
4) We suggested our limits in this experiment in the Discussion section as follows.
Thank you very much.
Yours sincerely,

Reviewer 2 Report
The manuscript describes the ameliorative effects of co-treatment of Gardeniae fructus and silymarin on liver fibrosis in TAA-induced mice model. The authors demonstrated sufficient evidence supporting their claims. There are some points could be revised:
1. How was Gardeniae fructus prepared? The procedures or steps for the preparation of GF could be indicated in the section of “Materials and Methods.”
2. The authors claimed that the effect of co-treatment of GF and silymarin was synergistic. However, the effects of co-treatment of drugs can be synergistic, additive, or antagonistic. There are some mathematical rules for the determination of these effects. Authors could calculate their co-treatment effect based on the mathematical approach.
Author Response
Comments and Suggestions for Authors
The manuscript describes the ameliorative effects of co-treatment of Gardeniae fructus and silymarin on liver fibrosis in TAA-induced mice model. The authors demonstrated sufficient evidence supporting their claims. There are some points could be revised:
Although the data may be of interest, there are numerous experimental flows that significantly diminish the enthusiasm and the quality for this manuscript.
â–º Thank you very much for the reviewer’s critical summary and important points of view. We tried to improve the present manuscript according to the reviewer’s comments.
- How was Gardeniae fructus prepared? The procedures or steps for the preparation of GF could be indicated in the section of “Materials and Methods.”.
â–º Thank you so much for your helpful comment. Actually, we prepared the GF with the same methods as we did previously; however, according to the reviewer’s instruction, we described it in the Material and Methods section.
- The authors claimed that the effect of co-treatment of GF and silymarin was synergistic. However, the effects of co-treatment of drugs can be synergistic, additive, or antagonistic.
There are some mathematical rules for the determination of these effects.
Authors could calculate their co-treatment effect based on the mathematical approach.
â–º We appreciate your special and professional comment. Actually, that is a very challenging issue in the current study. We previously calculated the combinations of concentration based on the two different cell lines which are HepG2 and LX-2 cells experiments. From these results, we found out the best concentration is 100 µg/mL to show anti-fibrotic effects on LX-2 cells without toxicity on the HepG2 cells by comparing each of the single dose treatments. To show the dose-dependent manners of GS by comparing to other groups, we selected these doses 50 and 100 µg/mL on the mice model.
Thank you very much.
Yours sincerely,
MR Shin, JA Lee and Prof. SS Roh
